# Development of Metabolic Syndrome Decreases Bone Mineral Density T-Score of Calcaneus in Foot in a Large Taiwanese Population Follow-Up Study

**DOI:** 10.3390/jpm11050439

**Published:** 2021-05-20

**Authors:** Hsuan Chiu, Mei-Yueh Lee, Pei-Yu Wu, Jiun-Chi Huang, Szu-Chia Chen

**Affiliations:** 1Department of General Medicine, Kaohsiung Chang Gung Memorial Hospital, Kaohsiung 833, Taiwan; mickey990055@gmail.com; 2Department of Internal Medicine, Division of Endocrinology and Metabolism, Kaohsiung Medical University Hospital, Kaohsiung Medical University, Kaohsiung 807, Taiwan; lovellelee@hotmail.com; 3Faculty of Medicine, College of Medicine, Kaohsiung Medical University, Kaohsiung 807, Taiwan; karajan77@gmail.com; 4Department of Internal Medicine, Kaohsiung Municipal Siaogang Hospital, Kaohsiung Medical University, Kaohsiung 812, Taiwan; wpuw17@gmail.com; 5Department of Internal Medicine, Division of Nephrology, Kaohsiung Medical University Hospital, Kaohsiung Medical University, Kaohsiung 807, Taiwan; 6Research Center for Environmental Medicine, Kaohsiung Medical University, Kaohsiung 807, Taiwan

**Keywords:** metabolic syndrome, osteoporosis, follow-up

## Abstract

Studies have suggested that there may be common pathogenic pathways linking osteoporosis and metabolic syndrome (MetS) due to the multiple risk factors for atherosclerotic cardiovascular disease caused by MetS. However, results on the association between MetS and bone health are inconsistent and sometimes contradictory. In this study, we aimed to investigate the associations between the effects of MetS risk factors and bone mineral density (BMD) T-score in a longitudinal study of 27,033 participants from the Taiwan Biobank with a follow-up period of 4 years. BMD of the calcaneus was measured in the non-dominant foot using ultrasound in the Taiwanese population. The overall prevalence rates of MetS were 16.7% (baseline) and 21.2% (follow-up). The participants were stratified into four groups according to the status of MetS (no/yes at baseline and follow-up). We investigated associations between MetS and its five components (baseline, follow-up) with BMD ΔT-score and found that the (no, yes) MetS group, (no, yes) abdominal obesity group, (no, yes) hypertriglyceridemia group, and (no, yes) low high-density lipoprotein (HDL) cholesterol group had the lowest ΔT-score. Furthermore, in the (no, yes) MetS group, high Δwaist circumference (*p* = 0.009), high Δtriglycerides (*p* = 0.004), low ΔHDL cholesterol (*p* = 0.034), and low Δsystolic blood pressure (*p* = 0.020) were significantly associated with low ΔT-score, but Δfasting glucose was not. In conclusion, in this large population-based cohort study, our data provide evidence that the development of MetS is strongly associated with increased rates of BMD loss in the Taiwanese population. This suggests that the prevention of MetS should be taken into consideration in the prevention of osteoporosis in the Taiwanese population.

## 1. Introduction

The prevalence of metabolic syndrome (MetS), a concurrence of abdominal obesity, glucose intolerance, hypertension, and dyslipidemia, is rapidly increasing globally, including in Taiwan. The prevalence of MetS (based on the NCEP-ATP III criteria) in Asian adults is estimated to range from 12% to 37% [1,2]. The recent Nutrition and Health Survey in Taiwanese adults estimated that the overall prevalence of MetS from 2013 to 2016 was 30.0% [3]. Individuals with MetS are at an increased risk of developing type 2 diabetes mellitus (DM) and atherosclerotic cardiovascular disease, and subsequently, a higher risk of cardiovascular disease-related and all-cause mortality [4,5,6].

Osteoporosis is also a global health problem and debilitating disease. It is associated with an increased risk of fractures and subsequent disability, leading to morbidity and mortality in the aging population [7,8]. Several studies have demonstrated that osteoporosis has common risk factors and pathophysiological mechanisms with atherosclerotic cardiovascular disease, such as smoking, low-grade inflammation, a sedentary lifestyle, increased oxidative stress, and sex hormone deficiency [9,10,11]. Other studies also have shown the possibility of common pathogenic pathways linking osteoporosis and MetS due to the multiple risk factors for atherosclerotic cardiovascular disease caused by MetS [12,13,14]. However, previous findings on the association between MetS and bone health have been inconsistent and sometimes contradictory [14,15,16,17].

Hormonal and biochemical profiles, as well as mechanical loading, are altered in patients with MetS, subsequently resulting in complex effects on bone metabolism. The proinflammatory state associated with MetS may result in a decrease in bone mass [18,19], and elevated blood pressure may increase urinary calcium excretion, thereby altering the mechanical loading, hormonal and biochemical profiles [20]. Increases in triglyceride and plasma glucose levels may also cause a low bone turnover and affect bone material properties [21,22]. In addition, microvascular complications associated with impaired glucose regulation may reduce blood flow to the bone [14]. On the other hand, obesity, overweight, and high body mass index (BMI) are known protective factors against excessive bone loss and are associated with a decreased fracture risk [23,24]. This may explain the controversial results in the relationship between MetS and bone mineral density (BMD). Due to these inconsistent results and the lack of a large cohort follow-up study on the association between MetS and BMD, the aims of this study were to investigate the associations among the combined effects of MetS risk factors and BMD T-score in a longitudinal study of 27,033 participants from the Taiwan Biobank (TWB).

## 2. Materials and Methods

### 2.1. Ethics Statement

This study was approved by the Institutional Review Board (IRB) of Kaohsiung Medical University Hospital (KMUHIRB-E(I)-20190398), and the TWB received ethical approval from the IRB on Biomedical Science Research, Academia Sinica, Taiwan, and the Ethics and Governance Council of the TWB. All of the participants provided written informed consent, and the study was conducted following the principles of the Declaration of Helsinki.

### 2.2. TWB

The TWB is the largest biobank supported by the Taiwanese government. The TWB was established with the aim of recording lifestyle and genomic data on Taiwanese residents [25,26]. The TWB includes data on community-based volunteers aged 30 to 70 years with no history of cancer. After signing informed consent, all participants provided blood samples and underwent face-to-face interviews and physical examinations. This study included 100,000 participants registered in the TWB, of whom 27,033 received follow-up examinations after a median of 4 years.

Recorded data included waist circumference (WC), body height and weight, and BMI was calculated as weight (kg)/height (m)^2^. During the face-to-face interviews, each participant completed a questionnaire with a TWB researcher that addressed personal information, lifestyle factors, diet, and personal and family medical histories. In this study, we defined regular exercise as participating in 30 min of “exercise” three times a week [27]. Further, we defined “exercise” as only including leisure activities such as yoga, jogging, swimming, hiking, cycling, computer-based dancing/exercise games/activities, playing a sport, etc. Physical work or heavy manual work related to occupational activities were not considered to be “exercise” in this study.

### 2.3. Collection of Demographic, Medical, and Laboratory Data 

Baseline variables including demographic features (age and sex), smoking history, medical history (DM and hypertension), examination findings (systolic (SBP) and diastolic blood pressure (DBP)), and laboratory data (fasting glucose, triglycerides, total cholesterol, high-density lipoprotein (HDL)-cholesterol, low-density lipoprotein (LDL)-cholesterol, hemoglobin, estimated glomerular filtration rate (eGFR) and uric acid) were recorded. EGFR was calculated using the 4-variable modification of diet in renal disease study equation [28].

### 2.4. Definition of MetS

Participants who met three of the following five criteria were defined as having MetS according to the NCEP-ATP III guidelines [29] and modified criteria for Asians [30]: (1) abdominal obesity, defined as WC >80 cm in women and >90 cm in men; (2) hypertriglyceridemia, defined as a triglyceride concentration ≥150 mg/dL; (3) low HDL cholesterol concentration, defined as <50 mg/dL in women and <40 mg/dL in men; (4) hyperglycemia, defined as a diagnosis of DM or a fasting whole-blood glucose concentration ≥110 mg/dL; and (5) high blood pressure, defined as SBP ≥ 130 mm Hg, DBP ≥85 mm Hg, a diagnosis of hypertension, or receiving treatment for hypertension.

### 2.5. Assessment of BMD

BMD (g/cm^2^) of the calcaneus was measured in the non-dominant foot using ultrasound (Achilles InSight, GE, USA). The T-score was calculated as follows: (the participant’s BMD—mean BMD in young adults)/standard deviation of a normal young-adult population.

### 2.6. Statistical Analysis

Data were expressed as percentage, mean ± standard deviation, median (25–75th percentile) for triglycerides, or mean ± standard error of the mean for BMD T-score. Multiple comparisons among the study groups were performed using one-way analysis of variance (ANOVA) followed by a Bonferroni-adjusted post hoc test. The study participants were stratified into four groups according to MetS status (no/yes at baseline and follow-up). Multivariable linear regression analysis was used to identify the determinants of BMD T-score and ΔT-score. The statuses (no, yes) of MetS, abdominal obesity, hypertriglyceridemia, low HDL cholesterol, and hyperglycemia, and high blood pressure were used as reference categories according to the lowest ΔT-score. A *p*-value of less than 0.05 was considered to indicate a statistically significant difference. All statistical analyses were performed using SPSS 19.0 for Windows (SPSS Inc. Chicago, USA).

## 3. Results

The mean age of the 27,033 participants (9555 males and 17,478 females) was 51.2 ± 10.4 years. The overall prevalence rates of MetS were 16.7% (baseline) and 21.2% (follow-up). The participants were stratified into four groups according to the status of MetS (no/yes at baseline and follow-up). Comparisons of the clinical characteristics among these study groups are shown in Table 1. There were 19,986, 2525, 1316, and 3206 participants in the four groups, respectively. Compared to the participants without MetS (no, no), those with MetS (yes, yes) were older, more predominantly male, and had higher prevalence rates of smoking, alcohol consumption, DM and hypertension, and higher SBP (baseline and follow-up), DBP (baseline and follow-up), BMI (baseline and follow-up), WC (baseline and follow-up), fasting glucose (baseline and follow-up), triglycerides (baseline and follow-up), hemoglobin (baseline and follow-up), uric acid (baseline and follow-up) and total cholesterol (baseline and follow-up). In addition, the participants with MetS (yes, yes) had lower HDL cholesterol (baseline and follow-up), LDL cholesterol (follow-up), and eGFR (baseline and follow-up). Moreover, the participants with MetS (yes, yes) had a lower BMD T-score (baseline and follow-up).

### 3.1. Associations among MetS and Its Five Components with Baseline BMD T-Score

Table 2 shows the associations among MetS and its components with baseline BMD T-score in the study participants. After adjusting for age, sex, smoking history, alcohol history, regular exercise, BMI, baseline laboratory findings including total cholesterol, LDL cholesterol, hemoglobin, eGFR, and uric acid, the participants with MetS (unstandardized coefficient β, −0.126; *p* < 0.001) were significantly associated with a low baseline T-score. Regarding each component of MetS, abdominal obesity (unstandardized coefficient β, −0.131; *p* < 0.001), hypertriglyceridemia (unstandardized coefficient β, −0.121; *p* < 0.001), low HDL cholesterol (unstandardized coefficient β, −0.087; *p* < 0.001), and high blood pressure (unstandardized coefficient β, −0.054; *p* = 0.016) were significantly associated with a low baseline T-score, but hyperglycemia was not.

Associations among MetS and its five components at baseline and follow-up with BMD ΔT-score.

Figure 1A–F illustrates the values of ΔT-score among MetS and its five components at baseline and follow-up.
MetSCompared to the (no, no), (yes, no), and (yes, yes) MetS groups, the (no, yes) MetS group had the lowest ΔT-score.Abdominal obesityCompared to the (no, no) and (yes, no) abdominal obesity groups, the (no, yes) abdominal obesity group had the lowest ΔT-score.HypertriglyceridemiaCompared to the (no, no), (yes, no), and (yes, yes) hypertriglyceridemia groups, the (no, yes) hypertriglyceridemia group had the lowest ΔT-score.Low HDL cholesterolCompared to the (no, no), (yes, no), and (yes, yes) low HDL cholesterol groups, the (no, yes) low HDL cholesterol group had the lowest ΔT-score.HyperglycemiaCompared to the (yes, yes) hyperglycemia group, the (no, yes) hyperglycemia group had a lower ΔT-score.High blood pressure

There was no statistically significant difference among the four groups.

The associations among MetS and its five components (baseline, follow-up) with BMD ΔT-score was further analyzed using multivariable linear regression analysis after adjusting for age, sex, smoking history, alcohol history, regular exercise, ΔBMI, Δtotal cholesterol, ΔLDL cholesterol, Δhemoglobin, ΔeGFR, and Δuric acid (Table 3).
MetSCompared to the (no, yes) MetS group, the (no, no) MetS group (unstandardized coefficient β, 0.043; *p* = 0.048), (yes, no) MetS group (unstandardized coefficient β, 0.101; *p* = 0.004), and (yes, yes) MetS group (unstandardized coefficient β, 0.091; *p* = 0.001) were significantly associated with high ΔT-score.Abdominal obesityCompared to the (no, yes) abdominal obesity group, the other three abdominal obesity groups were not significantly associated with ΔT-score.HypertriglyceridemiaCompared to the (no, yes) hypertriglyceridemia group, the (no, no) hypertriglyceridemia group (unstandardized coefficient β, 0.085; *p* < 0.001), (yes, no) hypertriglyceridemia group (unstandardized coefficient β, 0.144; *p* < 0.001), and (yes, yes) hypertriglyceridemia group (unstandardized coefficient β, 0.088; *p* = 0.001) were significantly associated with high ΔT-score.Low HDL cholesterolCompared to the (no, yes) low HDL cholesterol group, the (no, no) low HDL cholesterol group (unstandardized coefficient β, 0.081; *p* < 0.001), (yes, no) low HDL cholesterol group (unstandardized coefficient β, 0.135; *p* < 0.001), and (yes, yes) low HDL cholesterol group (unstandardized coefficient β, 0.112; *p* < 0.001) were significantly associated with high ΔT-score.HyperglycemiaCompared to the (no, yes) hyperglycemia group, the (yes, no) hyperglycemia group (unstandardized coefficient β, 0.118; *p* = 0.033) and (yes, yes) hyperglycemia group (unstandardized coefficient β, 0.106; *p* = 0.002) were significantly associated with high ΔT-score.High blood pressureCompared to the (yes, no) high blood pressure group, the (yes, yes) high blood pressure group (unstandardized coefficient β, 0.076; *p* = 0.025) was significantly associated with high ΔT-score.


Because menopause may have a crucial impact on the development of bone loss and osteoporosis, we further performed an analysis after excluding women who experience menopause at the follow-up period (*n* = 1869). The values of △T-score among MetS (no, no), MetS (no, yes), MetS (yes, no), and MetS (yes, yes) were −0.23, −0.29, −0.18, and −0.21, respectively (ANOVA *p* = 0.006). The lowest △T-score was noted in the group of MetS (no, yes). Furthermore, compared to the (no, yes) MetS group, the (yes, no) MetS group (unstandardized coefficient β, 0.059; *p* = 0.042), and the (yes, yes) MetS group (unstandardized coefficient β, 0.041; *p* = 0.036) were significantly associated with high ΔT-score. However, the (no, no) MetS group (unstandardized coefficient β, −0.036; *p* = 0.102) did not achieve significance.

### 3.2. Associations among ΔMetS Components with BMD ΔT-Score Using Multivariable Linear Regression Analysis in the (No, Yes) MetS Group

Table 4 shows the associations among ΔMetS components with ΔT-score in the study participants. After adjusting for age, sex, smoking history, alcohol history, regular exercise, ΔBMI, Δtotal cholesterol, ΔLDL cholesterol, Δhemoglobin, ΔeGFR, and Δuric acid, high ΔWC (per 1 cm; unstandardized coefficient β, −0.009; *p* = 0.009), high Δtriglycerides (per 10 mg/dL; unstandardized coefficient β, −0.008; *p* = 0.004), low ΔHDL cholesterol (per 1 mg/dL; unstandardized coefficient β, 0.006; *p* = 0.034), and low ΔSBP (per 1 mmHg; unstandardized coefficient β, 0.003; *p* = 0.020) were significantly associated with low ΔT-score, but Δfasting glucose was not.

## 4. Discussion

In this longitudinal analysis, we investigated the associations among MetS and its five components at baseline and follow-up with BMD among 27,033 participants who were followed up for a median of 4 years in the Taiwanese population. Overall, we found that after adjusting for confounders, the participants without MetS at baseline but with MetS at follow-up (the (no, yes) MetS group) had the highest rates of bone loss among the four groups. In addition, a similar association was revealed in the (no, yes) hyperglycemia, (no, yes) hypertriglyceridemia, and (no, yes) low HDL cholesterol groups, but not in the (no, yes) abdominal obesity or (no, yes) high blood pressure groups.

The first important finding of this study is that from the baseline cross-sectional data, we observed a negative correlation between MetS and BMD T-score. In addition, among the five MetS components, abdominal obesity, hypertriglyceridemia, low HDL cholesterol, and high blood pressure were associated with low T-score, whereas hyperglycemia was not. Although associations between MetS and BMD have been extensively studied, previous studies have been inconclusive with regards to the effect of MetS on bone. A meta-analysis by Xue et al. reported a significant association between an increased BMD of the spine and MetS in a total of 11 studies including 13,122 participants [31]. In addition, a meta-analysis conducted by Sugimoto et al. reported a possible association between MetS and bone loss in Asian men [32], and another meta-analysis by Zhou et al. of nine studies revealed a significantly lower BMD in men but not women [15]. Chronic low-grade inflammation and induction of oxidative stress are the hallmark features in the pathogenesis of MetS [33]. Proinflammatory cytokines and free radical species are known to modulate the bone remodeling process, particularly bone loss [34,35]. The inflammatory condition inhibits osteoblast-specific gene transcription and leads to the massive upregulation of downstream signaling in osteoclasts [36]. In addition, excessive oxidative stress promotes osteoclast differentiation and osteoblast and osteocyte apoptosis but suppresses osteoclast apoptosis and osteoblast differentiation, thereby affecting bone homeostasis [37]. We could not determine causality between MetS and low BMD owing to the cross-sectional nature of this study, and a longitudinal study would be more appropriate for this purpose.

The relationship between MetS and BMD is inconclusive among different ethnicities. In the Korea National Health and Nutrition Examination Survey, BMD of men with MetS was lower compared to those without MetS, but no difference was observed in women [38]. Rendina et al. showed a significant association between osteoporosis and MetS in European Caucasian women [39]. Other studies demonstrated a positive relationship between MetS and BMD. In the Rancho Bernardo study, MetS was associated with higher BMD in an American Caucasian cohort [12]. In the Third National Health and Nutrition Examination Survey III, a positive relationship between MetS and femoral neck BMD was observed in adult Americans [40]. In a homogenous Arab ethnic group, Wani et al. reported a significant positive association of MetS with BMD spine, and this association was independent of sex [41]. Chin et al. also demonstrated MetS is associated with increased BMD in the Malaysian population [42]. The present study is limited to the population of the Taiwan region. Therefore, our findings of the negative correlation between MetS and BMD T-score may not be generalizable to other ethnicities.

The second important finding of this study is that the (no, yes) MetS group had the greatest decrease in T-score. Moreover, the (no, yes) hypertriglyceridemia group, (no, yes) low HDL cholesterol group, and (no, yes) hyperglycemia group were also associated with the greatest decrease in T-score. However, there were no significant associations between BMD and the (no, yes) abdominal obesity or (no, yes) high blood pressure groups. MetS and its components had the lowest ΔT-score in (no, yes) groups and the highest ΔT-score in (yes, no) groups. This may suggest the improvement or deterioration of MetS more directly related to bone loss. MetS is a heterogeneous syndrome consisting of several disorders that interact with each other, and the mechanism behind the effects of MetS on bone metabolism is complicated and has yet to be fully elucidated. Therefore, investigations focusing on the effects of its individual components seem to be more appropriate in the context of bone metabolism. The mean age of the participants in this study is 51.2 years, and the average follow-up time is 4 years. It is important to note that many women experience menopause at this age, which has a crucial impact on the development of bone loss and osteoporosis. It has been proved that menopause is a major risk of BMD loss [43]. Therefore, we further performed analysis after excluding women who experience menopause at the follow-up period and found similar results that the (no, yes) MetS group had the greatest decrease in T-score.

The third important finding of this study was revealed when we considered each component of MetS separately as an independent variable and was that a high ΔWC was significantly and consistently associated with bone loss. Obesity is traditionally considered to protect against osteoporosis, as obese individuals have an increased body mass that exerts mechanical loading on the skeleton, thereby stimulating bone accrual [44]. Previous studies have also reported a positive association between central obesity and higher BMD at all sites in both genders [24]. However, other investigations have reported paradoxical outcomes, in that excess fat mass and visceral fat accumulation in obese subjects led to an increased risk of osteoporosis [45,46,47], which is consistent with the results of the present study. Obesity is associated with chronic low-grade inflammation and the overproduction of various proinflammatory cytokines in the systemic circulation, which can then stimulate osteoclast differentiation and bone resorption, subsequently resulting in loss of bone mass [48,49].

Another important finding of this study is that high Δtriglycerides and low ΔHDL cholesterol were associated with greater decreases in T-score. Dyslipidemia and obesity may share similar mechanisms with regards to osteoporosis because adipose tissue is a common feature of both conditions. Our results showed both a negative correlation between hypertriglyceridemia and BMD and also a great decrease in T-score in the subjects who developed hypertriglyceridemia. Lipid metabolism disorders can lead to high oxidized lipid levels, and lipid oxidization can stimulate adipocyte differentiation in bone marrow while suppressing osteoblast differentiation [50]. However, the relationship between dyslipidemia and osteoporosis is still unclear. Kim et al. found that triglyceride levels were negatively associated with BMD in postmenopausal women [51], and Wani et al. reported that hypertriglyceridemia resulted in an increase in the tertiles of T-score in Arab adults [36]. On the other hand, Loke et al. reported no significant correlation between triglycerides and BMD in an elderly Taiwanese population [17]. In addition, Adami et al. reported a positive correlation between triglycerides and BMD [52]. In addition, low HDL cholesterol was negatively associated with BMD, which is consistent with previous studies [17,53]. Moreover, we found that a low HDL cholesterol level was associated with a greater decrease in BMD. HDL cholesterol can prevent the toxic effects of oxidized LDL on osteoblastic cells by preserving lysosome integrity to inhibit oxidized LDL-induced apoptosis [54]. Moreover, HDL-associated paraoxonase 1 has been shown to suppress the formation of lipoproteins and oxidized lipids, thereby contributing to the protective effect of HDL cholesterol on bone [55]. In addition to cholesterol transport, HDL cholesterol also prevents the synthesis of proinflammatory cytokines from macrophages, which would otherwise favor the production of osteoclasts over osteoblasts, leading to bone loss [56].

The last important finding of this study is that low ΔSBP was associated with a great decrease in T-score. Abnormal calcium metabolism is a key link between hypertension and osteoporosis. Hypertension is associated with high levels of circulating sodium ions that prevent the reabsorption of calcium, which then increases the loss of calcium in urine [20,57]. In addition, removing calcium from the body through urination decreases the level of circulating calcium, resulting in the activation of parathyroid hormone and leading to a negative calcium balance on bone remodeling [58]. Theoretically, hypertension may be related to low bone mass due to changes in serum intact parathyroid hormone concentration and urinary calcium excretion; however, the results are controversial. Our results showed that higher blood pressure was significantly associated with BMD. Hanley et al. reported an association between hypertension and higher BMD in both genders [59]. However, Tseng et al. showed no significant association between systolic blood pressure and BMD in males or females, although they did find a strong inverse relationship between diastolic blood pressure and bone mineral loss in both genders [60]. Yang et al. reported that women with hypertension had lower femoral neck BMD, whereas men with hypertension were associated with higher femoral neck BMD [61].

Hyperglycemia has been shown to lead to bone loss through increased inflammatory responses and calcium metabolism disorders [62]. People with MetS have impaired glucose tolerance and are at a higher risk of developing type 2 DM [63]. Several previous studies have reported associations between impaired glucose tolerance and lower BMD compared to controls [41,64,65]. In addition, previous studies have reported an association between type 2 DM and decreased BMD and a higher risk of fractures [40,66]. We found that although there was no significant correlation between increased fasting glucose and baseline BMD, the development of hyperglycemia was associated with a marked increase in bone loss. Chronic hyperglycemia triggers the formation of advanced glycation end products when the carbonyl group of a reducing sugar reacts with the amino group of macromolecules [67]. The accumulation of advanced glycation end products can then induce osteoblast apoptosis through endoplasmic reticulum stress, leading to inferior bone quality and strength [68]. No significant association between hyperglycemia and baseline BMD T-score may be partly explained by lacking anti-diabetic agents. Anti-diabetic agents undoubtedly influence the values of fasting glucose, which may have an impact on the association between hyperglycemia and BMD T-score.

Further analyses of the (no, yes) MetS group revealed that the rate of BMD loss was much greater in the subjects with increased ΔWC or Δtriglycerides and that the subjects with high ΔHDL cholesterol or ΔSBP had markedly lower rates of bone loss. Surprisingly, the major discrepancy between the cross-sectional and longitudinal data was in the relationship between SBP and BMD. The presence of a high SBP level or hypertension was associated with lower baseline BMD in the cross-sectional data. However, in longitudinal data, the higher SBP seemed to be a protective factor against bone loss. This observation is in contrast to the findings of a previous meta-analysis [69], in which essential hypertension was associated with a significant reduction in BMD and positively associated with an increased risk of osteoporosis. It is generally thought that high blood pressure increases the excretion of calcium, resulting in an increase in parathyroid hormone level and subsequent bone resorption. Our finding of the lower BMD loss in the participants with a higher SBP is difficult to explain. A previous study demonstrated that SBP was strongly correlated with body size, and in particular skeletal muscle mass. Moreover, increased body size was shown to subsequently enhance peripheral resistance of the vascular system [70]. Therefore, we hypothesize that an elevated SBP may reflect an increased body size and possibly explain the positive association with BMD.

In this study, osteoporosis was confirmed using an Achilles InSight ultrasound device instead of dual-energy X-ray absorptiometry (DXA). The Achilles InSight has been shown to be able to identify osteoporosis defined by axial BMD using DXA in Chinese women [71]. The quantitative ultrasound (QUS) T-score at the left heel was positively correlated with the DXA T-score at the right heel (*r* = 0.90, *p* < 0.001) in 80 women, aged 53 to 73 years, with osteoporosis and/or fractures were followed repeatedly during 7 years [72]. Furthermore, QUS may be an improved predictor of fractures in comparison with DXA [73,74,75,76]. Although the most widely validated technique to measure BMD is DXA, and diagnostic criteria based on the T-score for BMD are a recommended entry criterion for the development of pharmaceutical interventions in osteoporosis. However, quantitative ultrasound has several advantages over DXA, including that radiation is not required, the low cost, and portability. Further studies are needed to assess eventually the correlation between QUS and the gold-standard technique for diagnosing osteoporosis based on DXA or quantitative computed tomography.

To the best of our knowledge, this is the first longitudinal study to examine the association between MetS and its components with BMD T-score. The main novelty and strength of the study are the large-scale investigation and follow-up of the association between MetS and its components with the BMD T-score. However, there are also several limitations to the present study. First, we lacked data on medication history, and certain medications may also be associated with the development or prevention of MetS or osteoporosis. No medication history, including anti-diabetic agents, antihypertensive medication, and lipid-lowering agents, was available in TWB, which had an impact on the longitudinal change of MetS. These medications undoubtedly influenced the values of fasting glucose, blood pressures, and lipid profile. Therefore, we could not exclude the impact of such medication on our present results. Second, although DXA is the standard method, ultrasound is a reasonable alternative when big populations are studied. However, just calcaneus BMD was measured. Different parts of the bone region were not measured, which may affect the ultimate results. Third, according to the statistics of TWB, the proportion of participants coming back to track is about 50%, which resulted in sample bias. Sampling bias is a systematic error due to a non-random sample of a population, in which all participants are not equally balanced or objectively represented. Therefore, sample bias may affect the interpretation of our results.

In conclusion, in this large population-based cohort study, our data provide dynamic evidence that improvement or deterioration of MetS more directly related to bone loss in the Taiwanese population. In addition, similar results were found in the subjects with hyperglycemia, hypertriglyceridemia, and low HDL cholesterol. This suggests that the prevention of MetS should be taken into consideration in the prevention of osteoporosis in the Taiwanese population.

## Figures and Tables

**Figure 1 jpm-11-00439-f001:**
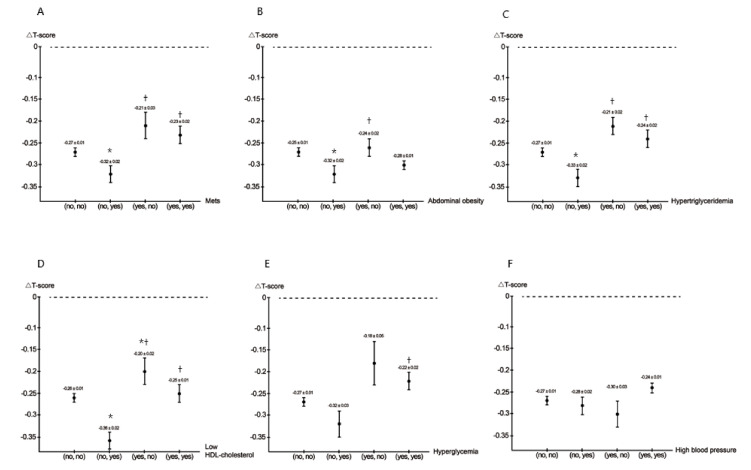
The values of △T-score among MetS (**A**), abdominal obesity (**B**), hypertriglyceridemia (**C**), low HDL cholesterol (**D**), hyperglycemia (**E**), and high blood pressure (**F**) (baseline, follow-up) groups. * *p* < 0.05 compared with (no, no); ^†^ *p* < 0.05 compared with (no, yes); ^#^ *p* < 0.05 compared with (yes, no).

**Table 1 jpm-11-00439-t001:** Comparison of clinical characteristics among study groups according to MetS status (baseline, follow-up).

Characteristics	MetS (No, No) (*n* = 19,986)	MetS (No, Yes) (*n* = 2525)	MetS (Yes, No) (*n* = 1316)	MetS (Yes, Yes) (*n* = 3206)
Age (year, baseline)	50.2 ± 10.4	52.7 ± 9.9 *	54.5 ± 9.4 *^,†^	54.9 ± 9.3 *^,†^
Age (year, follow-up)	54.0 ± 10.4	56.7 ± 9.8 *	58.2 ± 9.4 *^,†^	58.7 ± 9.3 *^,†^
Male sex (%)	33.8	37.9 *	42.2 *^,†^	40.0 *^,†,#^
Smoking (%)	23.8	29.4 *	30.9 *	31.7 *
Alcohol (%)	2.6	3.9 *	3.3	4.0 *
Regular exercise habits (%)	48.3	46.8	54.4 *^,†^	46.8 ^#^
Diabetes mellitus (%)	1.9	6.3 *	10.2 *^,†^	23.6 *^,†,#^
Hypertension (%)	7.2	17.3 *	25.2 *^,†^	41.7 *^,†,#^
SBP (mmHg, baseline)	114.0 ± 16.3	122.8 ± 16.2 *	128.3 ± 17.7 *^,†^	131.0 ± 17.4 *^,†,#^
SBP (mmHg, follow-up)	120.4 ± 17.6	134.2 ± 17.3 *	130.8 ± 17.8 *^,†^	136.7 ± 18.2 *^,†,#^
DBP (mmHg, baseline)	70.7 ± 10.2	75.3 ± 10.2 *	78.5 ± 11.0 *^,†^	79.2 ± 11.1 *^,†^
DBP (mmHg, follow-up)	72.0 ± 10.3	79.4 ± 10.6 *	76.9 ± 10.8 *^,†^	79.5 ± 11.3 *^,†^
BMI (kg/m^2^, baseline)	23.2 ± 3.1	25.8 ± 3.3 *	26.2 ± 3.4 *^,†^	27.6 ± 3.7 *^,†,#^
BMI (kg/m^2^, follow-up)	23.4 ± 3.2	26.5 ± 3.5 *	25.8 ± 3.3 *^,†^	27.7 ± 3.8 *^,†,#^
WC (cm, baseline)	80.6 ± 8.7	87.4 ± 8.8 *	89.8 ± 8.1 *^,†^	92.9 ± 9.0 *^,†,#^
WC (cm, follow-up)	81.4 ± 8.9	90.5 ± 8.6 *	87.6 ± 8.6 *^,†^	89.3 ± 9.1 *^,†,#^
Laboratory parameters				
Fasting glucose (mg/dL, baseline)	92.5 ± 12.8	99.0 ± 21.5 *	103.9 ± 25.4 *^,†^	113.8 ± 37.7 *^,†,#^
Fasting glucose (mg/dL, follow-up)	93.1 ± 13.3	104.3 ± 27.7 *	101.8 ± 25.4 *^,†^	116.7 ± 38.3 *^,†,#^
Triglyceride (mg/dL, baseline)	92.2 ± 53.3	130.9 ± 71.9 *	180.7 ± 83.4 *^,†^	209.3 ± 139.7 *^,†,#^
Triglyceride (mg/dL, follow-up)	95.6 ± 53.3	187.4 ± 106.0 *	128.3 ± 67.3 *^,†^	216.4 ± 186.3 *^,†,#^
Total cholesterol (mg/dL, baseline)	194.5 ± 34.5	198.4 ± 35.8 *	200.5 ± 36.3 *	196.7 ± 39.4 *^,#^
Total cholesterol (mg/dL, follow-up)	196.9 ± 35.3	198.3 ± 37.5	194.9 ± 37.0 ^†^	190.6 ± 40.1 *^,†,#^
HDL cholesterol (mg/dL, baseline)	57.5 ±12.8	48.8 ± 9.8 *	44.6 ± 8.8 *^,†^	42.1 ± 8.3 *^,†,#^
HDL cholesterol (mg/dL, follow-up)	58.0 ± 13.0	44.7 ± 9.0 *	49.4 ± 9.8 *^,†^	42.3 ± 8.6 *^,†,#^
LDL cholesterol (mg/dL, baseline)	120.7 ± 30.9	128.5 ± 32.6 *	125.8 ± 32.1 *	120.5 ± 34.1 ^†,#^
LDL cholesterol (mg/dL, follow-up)	120.7 ± 31.0	122.7 ± 33.2 *	122.9 ± 34.0	113.7 ± 34.5 *^,†,#^
Hemoglobin (g/dL, baseline)	13.6 ± 1.5	13.9 ± 1.6 *	14.1 ± 1.6 *^,†^	14.2 ± 1.6 *^,†^
Hemoglobin (g/dL, follow-up)	13.6 ± 1.5	14.0 ± 1.5 *	14.0 ± 1.5 *	14.1 ± 1.6 *
eGFR (mL/min/1.73 m^2^, baseline)	110.4 ± 24.8	106.6 ± 25.9 *	104.8 ± 25.2 *	105.1 ± 28.0 *
eGFR (mL/min/1.73 m^2^, follow-up)	108.1 ± 24.4	103.6 ± 25.7 *	103.0 ± 25.9 *	100.9 ± 29.7 *^,†^
Uric acid (mg/dL, baseline)	5.3 ± 1.3	5.9 ± 1.4 *	6.0 ± 1.4 *^,†^	6.3 ± 1.5 *^,†,#^
Uric acid (mg/dL, follow-up)	5.2 ± 1.3	6.0 ± 1.4 *	5.8 ± 1.4 *^,†^	6.1 ± 1.5 *^,†,#^
BMD T-score (baseline)	−0.44 ± 0.01	−0.50 ± 0.03	−0.63 ± 0.04 *	−0.65 ± 0.03 *^,†^
BMD T-score (follow-up)	−0.71 ± 0.01	−0.82 ± 0.03 *	−0.84 ± 0.04 *	−0.88 ± 0.03 *

Abbreviations. MetS, metabolic syndrome; SBP, systolic blood pressure; DBP, diastolic blood pressure; BMI, body mass index; WC, waist circumference; HDL, high-density lipoprotein; LDL, low-density lipoprotein; eGFR, estimated glomerular filtration rate. The study patients were stratified into 4 groups according to quartiles of MetS status (baseline, follow-up). * *p* < 0.05 compared with MetS (no, no); ^†^
*p* < 0.05 compared with MetS (no, yes); ^#^
*p* < 0.05 compared with MetS (yes, no).

**Table 2 jpm-11-00439-t002:** Association between MetS and its components with baseline BMD T-score using multivariable linear regression analysis in all participants.

Parameter	Multivariable	
	Unstandardized Coefficient β (95% CI)	*p*
MetS	−0.126 (−0.179, −0.074)	<0.001
MetS component		
Abdominal obesity	−0.131 (−0.178, −0.084)	<0.001
Hypertriglyceridemia	−0.121 (−0.170, −0.072)	<0.001
Low HDL cholesterol	−0.087 (−0.130, −0.044)	<0.001
Hyperglycemia	0.038 (−0.022, 0.098)	0.214
High blood pressure	−0.054 (−0.098, −0.010)	0.016

Values expressed as unstandardized coefficient β and 95% confidence interval (CI). Abbreviations are the same as in Table 1. Adjusted for age, sex, smoking history, alcohol history, regular exercise habit, BMI, baseline blood exams, including total cholesterol, LDL cholesterol, hemoglobin, eGFR, and uric acid.

**Table 3 jpm-11-00439-t003:** Association between MetS and its five components (baseline, follow-up) with BMD △T-score using multivariable linear regression analysis in participants.

MetS and Its Component	Multivariable	
	Unstandardized Coefficient β (95% CI)	*p*
MetS (no, no)	0.043 (0, 0.085)	0.048
MetS (no, yes)	Reference	
MetS (yes, no)	0.101 (0.033, 0.169)	0.004
MetS (yes, yes)	0.091 (0.038, 0.144)	0.001
Abdominal obesity (no, no)	0.023 (−0.017, 0.063)	0.268
Abdominal obesity (no, yes)	Reference	
Abdominal obesity (yes, no)	0.048 (−0.009, 0.105)	0.097
Abdominal obesity (yes, yes)	0.040 (0, 0.082)	0.053
Hypertriglyceridemia (no, no)	0.085 (0.044, 0.126)	<0.001
Hypertriglyceridemia (no, yes)	Reference	
Hypertriglyceridemia (yes, no)	0.144 (0.084, 0.205)	<0.001
Hypertriglyceridemia (yes, yes)	0.088 (0.037, 0.139)	0.001
Low HDL cholesterol (no, no)	0.081 (0.037, 0.125)	<0.001
Low HDL cholesterol (no, yes)	Reference	
Low HDL cholesterol (yes, no)	0.135 (0.076, 0.194)	<0.001
Low HDL cholesterol (yes, yes)	0.112 (0.062, 0.162)	<0.001
Hyperglycemia (no, no)	0.053 (−0.004, 0.110)	0.070
Hyperglycemia (no, yes)	Reference	
Hyperglycemia (yes, no)	0.118 (0.010, 0.227)	0.033
Hyperglycemia (yes, yes)	0.106 (0.038, 0.174)	0.002
High blood pressure (no, no)	0.036 (−0.028, 0.100)	0.267
High blood pressure (no, yes)	0.042 (−0.028, 0.111)	0.241
High blood pressure (yes, no)	Reference	
High blood pressure (yes, yes)	0.076 (0.009, 0.142)	0.025

Values expressed as unstandardized coefficient β and 95% confidence interval (CI). Abbreviations are the same as in Table 1. Adjusted for age, sex, smoking history, alcohol history, regular exercise habit, △BMI, △total cholesterol, △LDL cholesterol, △hemoglobin, △eGFR, and △uric acid.

**Table 4 jpm-11-00439-t004:** Association of △MetS components with BMD △T-score using multivariable linear regression analysis in participants with MetS (no, yes) (*n* = 2525).

Parameter	Multivariable	
	Unstandardized Coefficient β (95% CI)	*p*
ΔWC (per 1 cm)	−0.009 (−0.016, −0.002)	0.009
ΔTG (per 10 mg/dL)	−0.008 (−0.014, −0.003)	0.004
ΔHDL cholesterol (per 1 mg/dL)	0.006 (0, 0.012)	0.034
Δfasting glucose (per 1 mg/dL)	−0.002 (−0.003, 0)	0.051
ΔSBP (per 1 mmHg)	0.003 (0, 0.005)	0.020

Values expressed as unstandardized coefficient β and 95% confidence interval (CI). Abbreviations are the same as in Table 1. Adjusted for age, sex, smoking history, alcohol history, regular exercise habit, △BMI, △total cholesterol, △LDL cholesterol, △hemoglobin, △eGFR, and △uric acid.

## Data Availability

The data underlying this study is from the Taiwan Biobank. Due to restrictions placed on the data by the Personal Information Protection Act of Taiwan, the minimal data set cannot be made publicly available. Data may be available upon request to interested researchers. Please send data requests to: Szu-Chia Chen, Division of Nephrology, Department of Internal Medicine, Kaohsiung Medical University Hospital, Kaohsiung Medical University.

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
