# Peer review of "Development of Metabolic Syndrome Decreases Bone Mineral Density T-Score of Calcaneus in Foot in a Large Taiwanese Population Follow-Up Study"

_jpm, 2021, doi:10.3390/jpm11050439_

Round 1

Reviewer 1 Report

The revised version of the manuscript meets the review comments, and the quality has been well improved.

Author Response

The revised version of the manuscript meets the review comments, and the quality has been well improved.

Ans: Thank you for your review to make our manuscript better.

Reviewer 2 Report

The authors should consider the followings:

  1. Since osteoclast activities can be influenced the inflammation of MetS (or DM), the authors may include some paragraphs, of such contribution to MetS relationship.
  2. Please give more information of the ethnicities background of the subjects, since the variation in ethnicity may vary the results.
  3. Other than ethnicities, the authors may address whether the different part of the bone region (i.e. femoral, spine) being measure, may affect the ultimate results.
  4. In Table 2, please give rationales for the lower association in hyperglycaemia.
  5. The authors described some limitations of the studies, however, the authors can elaborate more on the limitations.
  6. Did the author only perform BMD measurement at calcaneus of the foot? If not, please explain why only measuring this region? Would the authors expect to see similar results on different bone regions?
  7. As a whole, the authors may adjust the tone of the article to address the scope of personalized medicine, i.e. by ethnicity specific or other means.
  8. The authors should mention the novelty of the article more clearly.
  9. The authors should revise the title of the article, to specifically include calcaneus in foot.

Author Response

The authors should consider the followings:

  1. Since osteoclast activities can be influenced the inflammation of MetS (or DM), the authors may include some paragraphs, of such contribution to MetS relationship. Ans: Thank you for your comments. We have added the issue in the Discussion.
  • Chronic low-grade inflammation and induction of oxidative stress are the hallmark features in the pathogenesis of MetS [33]. Proinflammatory cytokines and free radical species are known to modulate the bone remodelling process, particularly bone loss [34,35]. The inflammatory condition inhibits osteoblast-specific gene transcription and leads to the massive upregulation of downstream signaling in osteoclasts [36]. In addition, excessive oxidative stress promotes osteoclast differentiation, osteoblast and osteocyte apoptosis but suppresses osteoclast apoptosis and osteoblast differentiation, thereby affecting bone homeostasis [37]. (Line 276-283)

  1. Please give more information of the ethnicities background of the subjects, since the variation in ethnicity may vary the results.

Ans: Thank you for your suggestion. We have added the sentence “in Taiwanese population” in many places in the article. In the third paragraph of Discussion, we also added the discussion of the association of MetS and BMD in other ethnicity.

  1. Other than ethnicities, the authors may address whether the different part of the bone region (i.e. femoral, spine) being measure, may affect the ultimate results.

Ans: Thank you for your comments. In our study, BMD of the calcaneus was measured in the non-dominant foot using ultrasound. Different parts of the bone region were not measured. We have added in the Limitation.

  • Second, although DXA is the standard method but ultrasound is a reasonable alternative when big populations are studied. However, just calcaneus BMD was measured. Different parts of the bone region were not measured, which may affect the ultimate results. (Line 427-430)

  1. In Table 2, please give rationales for the lower association in hyperglycaemia.

Ans: Thank you for your comments. We tried to explain the results. In our study, we found that although there was no significant correlation between increased fasting glucose and baseline BMD, the development of hyperglycemia was associated with a marked increase in bone loss. We think that no medication history of TWB may influence the results. We have added the issue in the Discussion and Limitation.

  • No significant association between hyperglycemia and baseline BMD T-score maybe partly explained by lacking anti-diabetic agents. Anti-diabetic agents undoubtedly influence the values of fasting glucose, which may have an impact on the association between hyperglycemia and BMD T-score. (Line 382-385)
  • No medication history, including anti-diabetic agents, anti-hypertensive medication and lipid lowering agents, was available in TWB, which had an impact on the longitudinal change of MetS. These medications undoubtedly influenced the values of fasting glucose, blood pressures, and lipid profile. Therefore, we could not exclude the impact of such medication on our present results. (Line 423-427)

  1. The authors described some limitations of the studies, however, the authors can elaborate more on the limitations.

Ans: Thank you for your comments. We have elaborated on the limitations.

  • First, we lacked data on medication history, and certain medications may also be associated with the development or prevention of MetS or osteoporosis. No medication history, including anti-diabetic agents, anti-hypertensive medication and lipid lowering agents, was available in TWB, which had an impact on the longitudinal change of MetS. These medications undoubtedly influenced the values of fasting glucose, blood pressures, and lipid profile. Therefore, we could not exclude the impact of such medication on our present results. Second, although DXA is the standard method but ultrasound is a reasonable alternative when big populations are studied. However, just calcaneus BMD was measured. Different parts of the bone region were not measured, which may affect the ultimate results. Third, according to the statistics of TWB, the proportion of participants coming back to track is about 50%, which resulted in sample bias. Sampling bias is systematic error due to a non-random sample of a population, in which all participants are not equally balanced or objectively represented. Therefore, sample bias may affect the interpretation of our results. (Line 421-434)

  1. Did the author only perform BMD measurement at calcaneus of the foot? If not, please explain why only measuring this region? Would the authors expect to see similar results on different bone regions?

Ans: Thank you for your comments. In our study, BMD was just measured was measured of the calcaneus in the non-dominant foot using ultrasound. Different parts of the bone region were not measured. We have added in the Limitation. Although, the most widely validated technique to measure BMD is DXA. However, quantitative ultrasound has several advantages over DXA, including that radiation is not required, the low cost and portability. Although DXA is the standard method but ultrasound is a reasonable alternative when big populations are studied. Because we did not perform other bone regions, we could not make a conclusion to expect to see similar results on different bone regions. However, previous study (Eur J Radiol 2010, 76, (2), 265-8) showed that the Achilles InSight has been shown to be able to identify osteoporosis defined by axial BMD using DXA in Chinese women. Therefore, we expect to see similar results on different bone regions. Further studies are needed to assess eventually the correlation between QUS and the gold standard technique for diagnosing osteoporosis, based on DXA or quantitative computed tomography.

  • Second, although DXA is the standard method but ultrasound is a reasonable alternative when big populations are studied. However, just calcaneus BMD was measured. Different parts of the bone region were not measured, which may affect the ultimate results. (Line 427-430)

  1. As a whole, the authors may adjust the tone of the article to address the scope of personalized medicine, i.e. by ethnicity specific or other means.

Ans: Thank you for your suggestion. We have added the sentence “in Taiwanese population” in many places in the article. In the third paragraph of Discussion, we also added the discussion of the association of MetS and BMD in other ethnicity.

  1. The authors should mention the novelty of the article more clearly.

Ans: Thank you for your suggestion. We had revised the sentences to show the novelty of the article more clearly.

  • To the best of our knowledge, this is the first longitudinal study to examine the association between MetS and its components with BMD T‐score. The main novelty and strength of the study are the large-scale investigation and follow-up of the association between MetS and its components with BMD T‐score. (Line 417-420)

  1. The authors should revise the title of the article, to specifically include calcaneus in foot.

Ans: Thank you for your suggestion. We have revised the title to “Development of metabolic syndrome decreases bone mineral density T-score of calcaneus in foot in a large Taiwanese population follow-up study”.

This manuscript is a resubmission of an earlier submission. The following is a list of the peer review reports and author responses from that submission.

Round 1

Reviewer 1 Report

This study illustrates the relationship between metabolic syndrome and BMD T score based on East Asian population database. It is of great value to emphasis the risk factors or potential triggers of bone loss or osteoporosis in the current aging population. This study has a large sample size, a reasonable integration of data and a readable text, suggestions are as follow:

  1. This study is based on population of Taiwan region, is there any potential difference from other regions regarding MetS/ BMD? As mentioned in discussion part, “our findings may not be generalizable to other ethnicities”, the possible reasons could be also discussed in this part.
  2. The definition “exercise” should be “at least participating in 30 minutes of exercise three times a week”?
  3. The mean age of the sample in this study is 51.2 years and the average follow-up time is 4 years. It is important to note that many women experience menopause at this age, which has a crucial impact on the development of bone loss and osteoporosis. This may dilute the effect of MetS on bone loss and osteoporosis. Therefore, are the results after selecting only male in the sample, or excluding postmenopausal women, consistent with the current results? This aspect could also be discussed in the discussion part.
  4. As figure 1 shown, MetS and its components always has the lowest ΔT-score in (no, yes) groups and highestΔT-score in (yes, no) groups. This may suggest the improvement or deterioration of Me'tS more directly related to bone loss? On the other hand, it is interesting that (no, no) group doesn’t have the highest ΔT-score especially compared with (yes, yes) group, which indicates that people with MetS all the time has higher BMD T score than people without MetS? Such dynamic analysis of the data will add value to this longitudinal study.
  1. As a large population-based longitudinal analysis, it is not sufficient to only conclude that MetS is related to BMD loss in the conclusion part. The findings from dynamic analysis should be mentioned as well.

Reviewer 2 Report

While this manuscript analyzes are large data set in relation to MetS, measuring BMD through a single variable (Calcaneus BMD) is not sufficient or relevant to overall BMD. Either further data inclusion of other BMD measurements should be analyzed, or the bone density aspect of this study should be removed.